# Elevation Multi-Channel Imbalance Calibration Method of Digital Beamforming Synthetic Aperture Radar

**Hao Chen [1,2], Feng Ming [1,*], Liang Li [1,2] and Guikun Liu [1,2]**

[1] National Key Laboratory of Microwave Imaging Technology, Aerospace Information Research Institute, Chinese Academy of Sciences, Beijing 100190, China

[2] School of Electronic, Electrical and Communication Engineering, University of Chinese Academy of Sciences, Beijing 100049, China

**\*** Correspondence: mingfeng@aircas.ac.cn

**Abstract:** The digital beamforming synthetic aperture radar (DBF-SAR) is proposed by scholars as a promising solution to overcome the constraint of the minimum antenna area of the traditional single-channel SAR to achieve high resolution and wide swath (HRWS) by scan-on-receive (SCORE) in the elevation multiple channel. However, the inevitable channel imbalance between the elevation channels of DBF-SAR will degrade the DBF-SAR image quality. In this paper, we present a method to estimate the sampling time delay error, amplitude error and phase error based on the external calibration data. For the sampling time delay deviation, we adopt to calculate the statistical average of the position deviation of several external calibration points in the reference channel image with that of the error channel image. To avoid noise interference, we image the DBF-SAR original echo-carrying amplitude information to obtain the amplitude error between channels by dividing the absolute values of the complex image data of the error channel. Due to the phase error between channels, the image contrast will decrease. Therefore, the problem of estimating the phase error can be transformed into the problem of maximizing the image contrast. So, in this paper, we use the gradient descent method to optimally estimate the phase error. Finally, the effectiveness of the method is verified by the simulation of airborne measured data and simulation data.

**Keywords:** SAR; digital beamforming; external calibration

## 1. Introduction

Synthetic aperture radar (SAR) is an active ground sensing system which might be used in severe weather conditions, such as clouds, rain and fog [1]. Throughout the development of SAR, high resolution and wide swath imaging have always been two important factors driving the development of SAR [2–4]. The high resolution can provide more detailed target information, which is good for target identification and feature extraction [5,6]. In future Earth observation applications, SAR is usually required to have continuous global coverage, such as wide-range map mapping, regular homeland applications, ocean and glacier monitoring, etc. [7,8]. Wide swath also means shortening the revisit period of the same area so that a specific area of the Earth's surface will be observed at a higher frequency for immediate analysis of the situation and problem solving [9]. Consequently, high resolution and wide swath (HRWS) have become the important indicators of future SAR imaging capability.

SAR with high resolution imaging of the ground (such as resolution of more than 1 m) and wide enough ground swath imaging (such as swath of more than 100 km), are more suitable for future remote sensing and telemetry applications [10]. This has made high resolution and wide swath imaging an important research direction in the field of SAR research in recent years.

Due to the existence of minimum antenna area, the contradiction between azimuth resolution and mapping width has become a difficult obstacle for traditional single-channel SAR systems to overcome [11]. High resolution and wide swath are two contradictory metrics that cannot be satisfied simultaneously. The solution to the contradiction between high resolution and wide swath of traditional single-channel SAR is to adopt multi-channel SAR systems.

The elevation multi-channel technology is a new technology established with the development of digital signal processing methods, which not only can completely retain the information collected on the array antenna, but also can utilize complicated digital signal processing methods to process the signal. It can form flexible transmitting and receiving beam outputs using digital signal processing techniques at the baseband digital end, which we call digital beamforming (DBF) [12].

DBF technology applies in the SAR receiver port, using the scan-on-receive (SCORE) technique [13], which divides the antenna at the receiver port into multiple sub-apertures, and the signals received by each sub-aperture are amplified, down-converted and digitized separately. As a result, the SCORE-based system increases the reception gain of the useful signal and suppresses the range ambiguity compared to the traditional single-channel SAR system.

Although the technical feasibility and performance of elevation multi-channel SCORE systems have been demonstrated in numerous studies over the past few years, the identification and validation of adequate calibration strategies are still lacking and urgently needed.

Several papers in recent years have proposed various SAR operating modes in conjunction with SCORE technology [14,15], and the research on elevation multi-channel SCORE technology has mainly focused on solving the elevation mismatch problem that exists due to ambiguous range and large terrain undulations [16,17], as well as on the optimization of the weighting coefficients for each channel in the elevation [18]. In the discussion of these problems, most of the articles assume that each channel is in an ideal state and do not consider the variability between channels in practice.

With the widening of the elevation swath, the slant range error between the ground target and each channel is spatially variable along the elevation, which makes it difficult to maintain high reception gain throughout the entire swath with the elevation multi-channel SCORE technique. The introduction of the multi-channel will bring a variety of errors, and in the actual working process, due to the influence of temperature, power supply voltage and system manufacturing process, the frequency response function of each channel does not maintain good consistency, which leads to the beam pointing deviation after DBF being very large [19].

Consequently, the multi-channel SAR systems need to be calibrated. However, the major existing calibration methods for multi-channel systems are for azimuthal multi-channel systems [20–22], as there are already on-orbit azimuthal multi-channel systems, such as GF-3 [23], and there is no on-orbit spaceborne DBF-SAR system yet, but some countries have developed airborne DBF-SAR systems to experiment for future on-orbit spaceborne DBF-SAR systems [24,25]. Up to now, for the azimuth multi-channel SAR system, the calibration methods fall into two main categories: one is the orthogonal subspace method, and the other method is the azimuthal correlation method [26–28]. However, due to the complex structure of DBF-SAR, the above calibration methods could not apply to DBF-SAR. Some internal calibration methods are proposed to solve the imbalance between elevation channels of DBF-SAR [29,30]. However, the internal instrument calibration methods exclude the error source in the SAR antenna. In the actual work, the SAR antenna as an error source must be considered.

So, in this paper, the echo model of each receiving channel and the model for DBF-SAR are demonstrated. Furthermore, we propose an external calibration-based DBF-SAR channel calibration method, which uses the strong scattering characteristics of the external calibration points. The method requires the multi-channel data of the external calibration

area to be downlinked to the ground for processing. This is based on the following considerations: (1) The longitude, latitude, elevation and corner reflector information of the external calibration area are all known. (2) The multi-channel data are firstly downlinked to the ground for processing, and then the estimated sampling delay, amplitude and phase errors are uploaded to the satellite and corrected, which can considerably reduce the computational workload on the satellite. (3) This means that it is feasible to correct the estimated inter-channel errors to the satellite in real time. (4) The ability to downlink data from each channel to the ground is necessary for DBF-SAR in system design.

This paper is organized as follows. In Section 2, the elevation multi-channel DBF-SAR is analyzed and modeled in detail. In Section 3, the proposed method is described in detail. In Section 4, the processing results for both simulated data and real data are presented and quantitatively analyzed. As a comparison, the results of the traditional methods are presented. Section 5 is the discussion. Section 6 presents the conclusions of the paper.

## 2. Model of DBF-SAR

The traditional single-channel SAR system adopts single transmitting and single receiving mode in operation. During the whole illuminating time and receiving time, the main lobe of the antenna pattern covers the whole mapping strip, and the beam points to the center of the mapping strip and does not change. Thus, the gain at the center of the mapping strip is maximal, while the gain at the edge of the mapping strip will be gradually reduced. However, the DBF-SAR adopts single transmitting and multiple receiving mode; Figure 1 shows the geometric schematic diagram of the working of the DBF-SAR. It can be seen from the figure that the receiving antenna is divided into multiple sub-apertures in elevation, and each sub-aperture can be regarded as a channel. One sub-aperture transmits a wide beam to cover the mapping strip when transmitting, and all sub-apertures transmit a wide beam to receive. By weighting the received signals of each sub-aperture and accumulating them, a time-varying high-gain narrow receiving beam can be formed, and the narrow beam can be used for real-time scanning reception, which can improve the reception gain, achieve the maximum gain reception of the echo signal for the whole swath, and reduce range ambiguity.

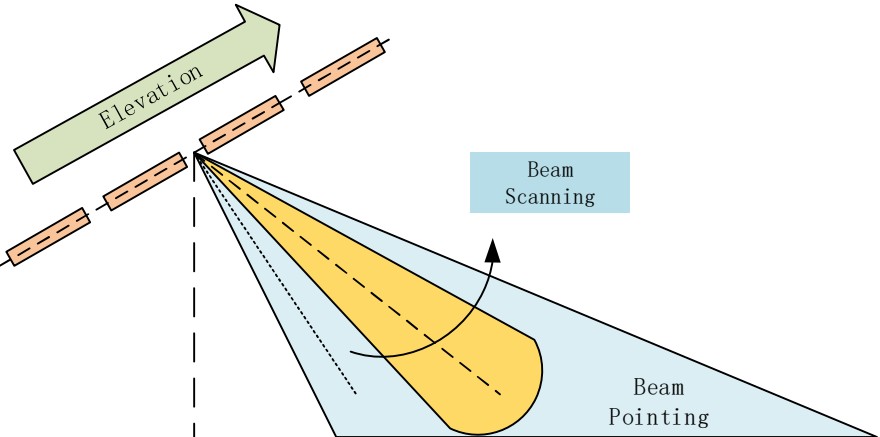

**Figure 1.** DBF geometric schematic diagram.

The block diagram is shown in Figure 2, which shows that the received signal is processed separately by each receiving channel, and then amplified, downconverted and I/Q quadrature demodulated into a zero IF signal without distortion. Then the zero IF signal

is transformed into a digital signal by A/D conversion, and finally the digital signal obtained from each channel is weighted and summed up to realize the DBF of the elevation multi-channel signal.

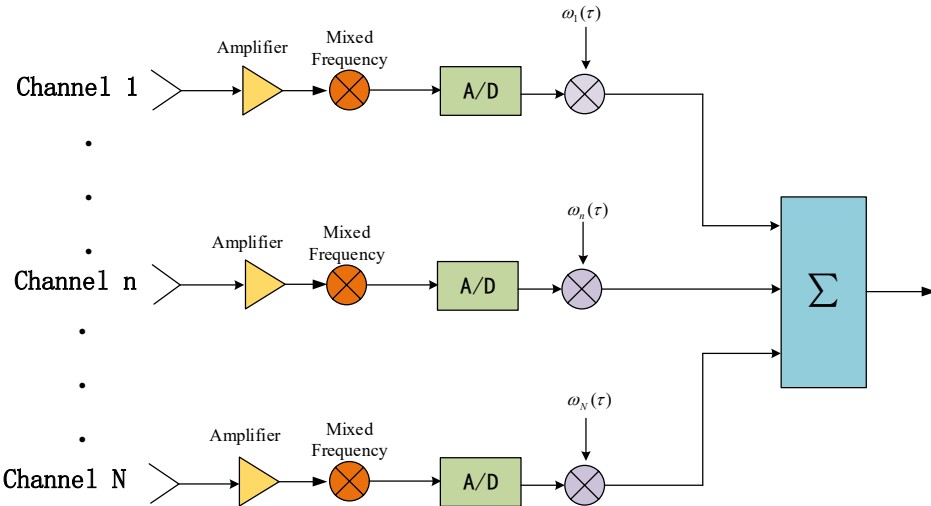

**Figure 2.** DBF block diagram.

Through multi-channel signals after DBF processing, echo energy enhancement can be achieved, extending the effective mapping bandwidth, generating a flexible high-gain narrow receive beam, and tracking the instantaneous arrival direction of the radar pulse echo in real time, which maximizes the receive gain of the signal of interest and effectively reduces ambiguous echoes.

The chirp signal transmitted by the radar transmitter can be expressed as

$$s(t) = \gamma \cdot rect \left| \frac{t}{T} \right| \cdot \exp\left\{ j\pi K_r t^2 + j2\pi f_c t \right\} \tag{1}$$

where $\gamma$ is denoted as a complex constant related to the backscattering and incident angle, $f_c$ is the carrier frequency, $K_r$ is the FM rate, $T$ is the pulse duration, and $t$ is the fast time.

Taking a point target as an example, the echo signal of the point target received by the reference channel is

$$s_1(t) = \gamma \cdot rect \left| \frac{t - t_0}{T} \right| \cdot \exp\left\{ j\pi K_r (t - t_0)^2 + j2\pi f_c (t - t_0) \right\} \tag{2}$$

The $t_0$ in the above equation represents the time from the point target to the reference channel after transmitting a pulse.

The echo signal received by the nth aperture is amplified, down-converted and digitized to obtain the digital baseband signal as

$$s_n(t) = \gamma \cdot rect \left| \frac{t - t_n}{T} \right| \cdot \exp\left\{ j\pi K_r (t - t_n)^2 - j2\pi f_c t_n \right\} \tag{3}$$

where $n = 1, \ldots, N$

$$t_n = t_0 - \frac{d_n \cdot \sin(\theta(t_0) - \beta)}{c} \tag{4}$$

where $d_n$ is the distance between the nth aperture and the reference channel, $\theta(t_0)$ is the beam look down angle at time $t_0$, and $\beta$ is the antenna normal look down angle. Let

$\alpha(t_0) = \theta(t_0) - \beta$, where $\alpha(t_0)$ is the antenna normal deviation angle, so $\alpha_0 = \alpha(t_0)$. Supposing the Earth is a sphere, the angle of view corresponding to the direction of echo received by radar at time $t$, $\theta(t)$ could be expressed as

$$\theta(t) = \arccos\left[\frac{(H+R_e)^2 + R^2(t) - R_e^2}{2 \cdot (H+R_e) \cdot R(t)}\right] \tag{5}$$

where $R_e$ is Earth's radius, and $R(t)$ represents the target slant range corresponding to the echo received at time $t$. The weight coefficient $\omega_n(t)$ could be represented as

$$\omega_n(t) = \exp\left(-j2\pi d_n \sin(\alpha(t))/\lambda\right) \tag{6}$$

The signal after weighting by the weight coefficient and summed up is

$$y_n(t) = \gamma_1 \cdot rect\left(\frac{t-t_n}{T}\right)\exp\left(j\pi k_r(t-t_n)^2 + j\frac{2\pi}{\lambda}d_n\left(\sin\alpha_0 - \sin(\alpha(t))\right)\right) \tag{7}$$

The echo range of the point target can be expressed as $2R_0/c - T/2 \leq t \leq 2R_0/c + T/2$ when the transmitting pulse passes the point target; the digital beamforming by the weighting process only points to the direction of the point target when $t_0 = 2R_0/c$, and thus the maximum gain of the antenna is not available at other echo times.

Therefore, the output signal after DBF could be expressed as

$$y(t) = \sum_{n=1}^{N}\omega_n(t) \cdot s_n(t) = \sum_{n=1}^{N}y_n(t) \tag{8}$$

## 3. Method

The above signal model is the ideal signal model. However, in the DBF-SAR system, each individual received channel contains a power amplifier, mixer, low-noise amplifier, filter, A/D converter and transmission cable, so there are small differences in the connection status of electronic devices among channels, resulting inevitably in channel errors, such as amplitude error, phase error, and sampling delay error among channels.

Usually, the slant range difference between the target in the mapping strip and the receiving antenna for each channel is tiny, commonly centimeters in DBF-SAR [31], so it can be assumed that the elevation signal envelope between each channel is the same. Therefore, the signal model with time delay, amplitude and phase errors received by the receiving channel can be approximated as

$$s_{n\_err}(t) = \Delta A_n \cdot \gamma \cdot rect\left|\frac{t-t_n-\Delta t_n}{T}\right| \cdot \exp\left\{j\pi K_r(t-t_n-\Delta t_n)^2 - j2\pi f_c t_n\right\} \cdot \exp\left\{j\Delta\phi_n\right\} \tag{9}$$

where $\Delta A_n$ is the amplitude error, $\Delta t_n$ is the sampling delay error, $\Delta\varphi_n$ is the phase error. When the channels exist above channel imbalance, it will result in the beam pointing after weighted and be summed to the incorrect main lobe gain decrease and sidelobe rise. Figure 3 shows the influence of the channel imbalance, where the red dotted line is the single channel antenna pattern and the blue line is the antenna pattern of DBF-SAR.

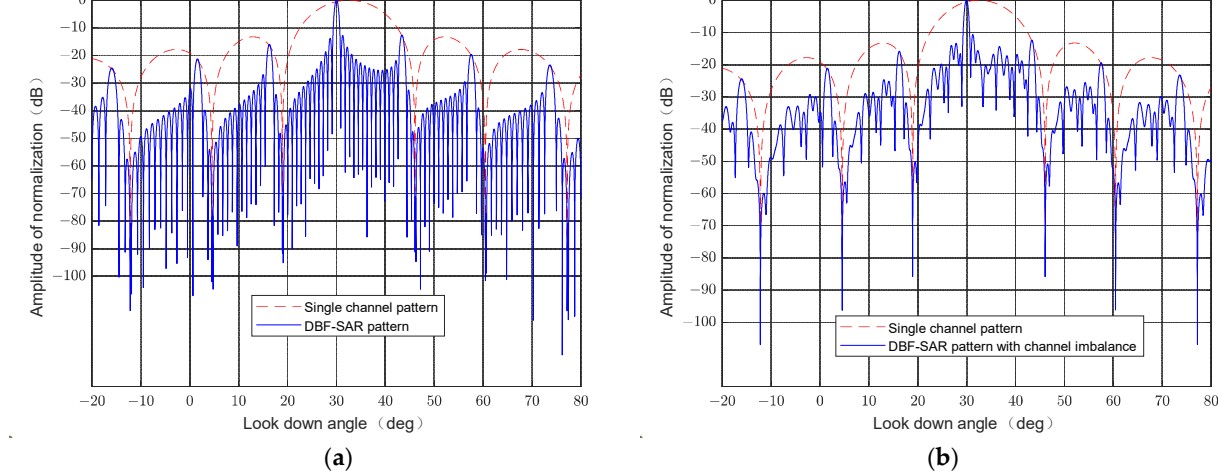

**Figure 3.** The effect of channel mismatch on the full array antenna pattern. (**a**) Without channel imbalance; (**b**) with channel imbalance.

We could obtain the SAR image of each channel by multi-channel data for calibration. The two-dimensional imaging model of multi-channel data received on the ground is

$$
\begin{aligned}
I_{n\_err}(t,\eta) = \Delta A_n \cdot \gamma \cdot \sin c\left(K_r T\left(t - t_0 - \Delta t_n\right)\right) \sin c\left(B_a \eta\right) \\
\cdot \exp\left(-j2\pi f_c t_0\right) \exp\left(j\frac{2\pi}{\lambda} \mathrm{d}_n \sin(\theta_0 - \beta)\right) \exp(j\Delta\phi_n) + n(t,\eta)
\end{aligned}
\tag{10}
$$

The channel imbalance in the DBF-SAR system will cause the deviation of the weighted beam pointing, the main lobe gain decreases and the sidelobe increases. Finally, it will decrease the DBF-SAR image quality.

This paper proposes a method based on the external calibration to solve the channel imbalance. So, for the purposes of our discussion, we need to downlink the multi-channel data to the ground for processing, which can significantly reduce the amount of in-orbit computing. Figure 4 shows the workflow of the proposed method.

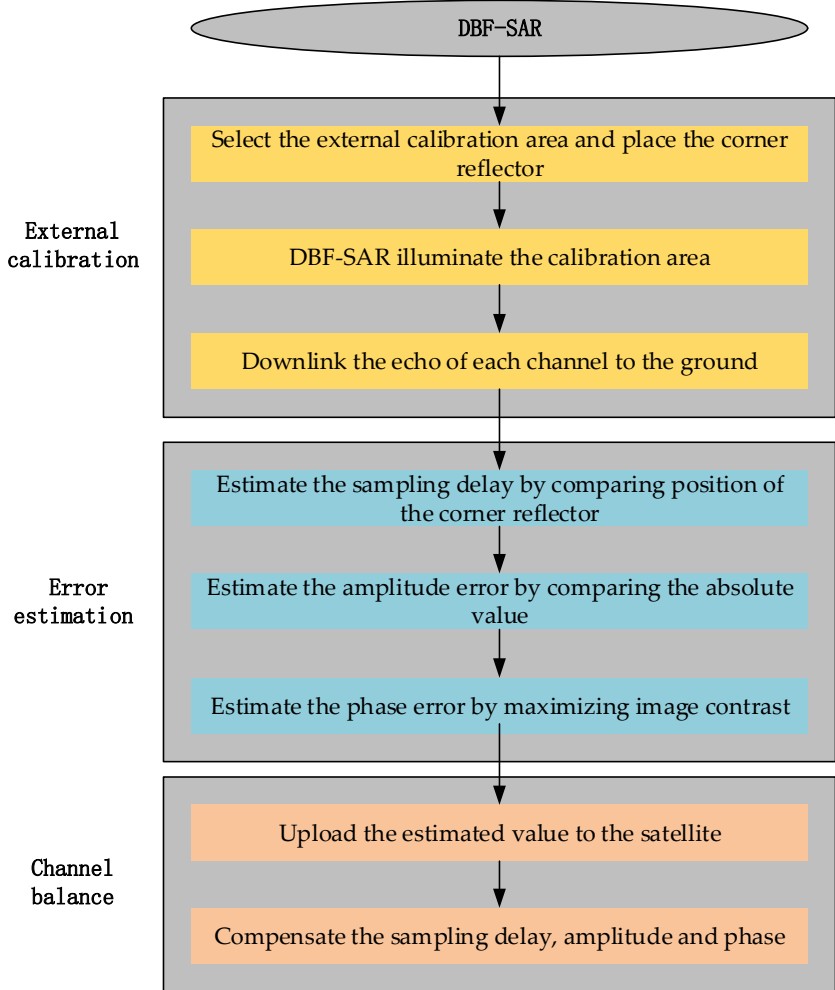

**Figure 4.** Workflow of the proposed method.

*3.1. Physical Experiment Conditions*

3.1.1. Selection of the Calibration Equipment

Commonly used SAR calibrators are divided into two types, active/passive [32]: active calibrators include receivers, transponders, while passive calibrators include the trihedral corner reflector, dihedral corner reflector, metal ball, etc.

The transponders can be designed with high radar cross section (RCS) values [33], so the device can be miniaturized, but at a high cost. Passive calibrators' RCS values are related to their size. In order to achieve the same RCS as transponders, the equipment is designed to be huge, but because of their lower cost, they are widely used for calibrations.

Considering the cost, RCS, deployment difficulty, and other factors, we choose the trihedral corner reflector here. The RCS value of the trihedral corner reflector should be 20 dB higher than the total emissivity of the SAR image resolution unit, but not more than the upper limit of the level signal that can be received. The formula for calculating the RCS of a trihedral corner reflector is as follows:

$$\sigma = \frac{4\pi a^4}{3\lambda^2} \tag{11}$$

where *a* is the side length of the trihedral corner reflector, and $\lambda$ is the wave length. So, we choose the trihedral corner reflector with the side length of approximately 0.7 m, for which the RCS is approximately 30 dB for the calibration.

### 3.1.2. Selection of the Calibration Field

The placement of the trihedral corner reflectors should consider the impact of the background clutter of the terrain. So, they should be placed in a flat and level area to avoid the influence of shadows and shading.

### 3.2. Sampling Delay Estimation Method

We adopt the external calibration method to estimate the sampling delay, so we put the corner reflector on the ground. By illuminating the corner reflector area, we could obtain the multi-channel data with information of the corner reflector. Then use the multi-channel data to obtain every receiving channel SAR image, where the ground control point shows as a peak point in the SAR image. Let the transmitted channel be the reference channel.

Through the geographic location of the ground control point and the pixel coordinates of the ground control point on the reference image, where the pixel coordinate of the i'th ground control point in the reference channel SAR image is $(x_{ref,i}, y_{ref,i})$ and the pixel coordinate of the i'th ground point in the n'th receive channel SAR image is $(x_{n,i}, y_{n,i})$, the sampling delay can be estimated by comparing the slant range of the reference channel and other error channels.

The sampling delay of each ground control point can be expressed as

$$\triangle delay_{n,i} = \sqrt{((x_{ref,i} - x_{n,i}) \cdot c/2)^2 + ((y_{ref,i} - y_{n,i}) \cdot V_r)^2} \tag{12}$$

The estimated value of sampling time delay error is the mean value of the sampling delay deviation of each control point of the channel. The sampling delay of each receiving channel contains only accidental errors. According to the statistical definition of truth, when the observed quantity contains only accidental errors, its mathematical expectation is its truth value. Therefore, the statistical average value of several fixed punctuation delays for each channel is the channel delay fixed value of the channel.

However, the point target in the SAR image is formed by multiple high amplitude pixels; comparing the peak point positions of the reference channel image and the error channel image directly will produce relatively large deviations. Therefore, we interpolate the local SAR image to determine the position of the corner reflector in the SAR image with an accuracy better than 1/10th of a pixel. When the range sampling distance is 1 m, the control point measurement accuracy in the SAR image is 0.1 pixel, and the error caused by the inaccurate measurement of control points is better than 0.1 m.

The steps are shown below:

1. To obtain the SAR image of each channel on the ground;
2. Obtain the pixel coordinates of all calibration points in each channel SAR image;
3. Calculate the position deviation of each calibration point in the reference channel image and the error channel image after interpolating, and calculate the mean value that is the sampling delay estimated result.

### 3.3. Amplitude Error Estimation Method

Here, we use the strong scattering of the corner reflector to estimate the amplitude error between channels, so the corner reflector is placed on a flat and level scene for calibration. In this case, the corner reflector is a strong target point in the scene and is represented as a peak point in the echo signal.

Because the distance between the receiving channels is tiny, with the echo signal after range compression, the *sinc* function nearly does not change near the maximum value. So, when there is no noise, the amplitude error between channels can be obtained by calculating the ratio of the reference channel signal and the error channel signal after range compression. The traditional method is to filter the noises of echoes of different channels and reference channels on the satellite first to avoid the interference of noise, and then to

divide and obtain the amplitude error of each channel. However, the noise is often very difficult to eliminate thoroughly, so in practice, we could consider obtaining the point target SAR image to reduce the effect of noise.

$$I_{ref}(t,\eta) = \gamma \cdot \sin c\left(K_r T\left(t-t_0\right)\right)\sin c\left(B_a\eta\right)\exp\left(-j2\pi f_c t_0\right) + n(t,\eta) \tag{13}$$

Therefore, the amplitude error could be expressed as

$$
\begin{aligned}
\Delta\hat{A}_n &= abs\left(\frac{\max(I_{n\_err}(t,\eta))\cdot\exp\left(-j\dfrac{2\pi}{\lambda}d_n\sin(\theta_0-\beta)\right)}{\max(I_{ref}(t,\eta))}\right)\\[2mm]
&= abs\left(\frac{\Delta A_n\cdot\gamma\cdot\sin c\left(K_r T\left(t-t_0\right)\right)\sin c\left(B_a\eta\right)\exp\left(-j2\pi f_c t_0\right)\exp\left(j\Delta\phi_n\right)+n(t,\eta)}{\gamma\cdot\sin c\left(K_r T\left(t-t_0\right)\right)\sin c\left(B_a\eta\right)\exp\left(-j2\pi f_c t_0\right)+n(t,\eta)}\right)\\[2mm]
&\approx abs(\Delta A_n\exp(j\Delta\phi_n))
\end{aligned}
\tag{14}
$$

The steps are shown below:

1.  To obtain SAR image of each channel on the ground;
2.  Calculate the amplitude of the complex image data of the reference channel and the error channel, and then divide them.

### 3.4. Phase Error Estimation Method

Due to the structure of DBF-SAR, it is known that, due to the different positions of different receiving apertures, i.e., each signal has phase error due to the different return distances, it is necessary to introduce the corresponding phase compensation factor $\exp\left(j\dfrac{2\pi}{\lambda}d_n\sin(\theta_0-\beta)\right)$ for each receiving channel, and then the phase estimation is performed. Therefore, after compensating for the fixed phase deviation, the phase error can be obtained by dividing the reference channel and the error channel after imaging and taking the phase angle.

$$
\begin{aligned}
\Delta\hat{\phi}_n &= \arg\left(\frac{\max(I_{n\_err}(t,\eta))\cdot\exp\left(-j\dfrac{2\pi}{\lambda}d_n\sin(\theta_0-\beta)\right)}{\max(I_{ref}(t,\eta))}\right)\\[2mm]
&= \arg\left(\frac{\Delta A_n\cdot\gamma\cdot\sin c\left(K_r T\left(t-t_0\right)\right)\sin c\left(B_a\eta\right)\exp\left(-j2\pi f_c t_0\right)\exp\left(j\Delta\phi_n\right)+n(t,\eta)}{\gamma\cdot\sin c\left(K_r T\left(t-t_0\right)\right)\sin c\left(B_a\eta\right)\exp\left(-j2\pi f_c t_0\right)+n(t,\eta)}\right)\\[2mm]
&\approx \arg(\Delta A_n\exp(j\Delta\phi_n))
\end{aligned}
\tag{15}
$$

However, due to many components of phase, the above method is commonly very inaccurate in extracting the phase error, so here we use an algorithm based on the maximum contrast of the image to estimate the phase error between channels [34].

Since the existence of the obtained phase error between channels decreases the image contrast between channels, the larger the phase error, the smaller the contrast, and the image contrast is maximized when there is no error. Therefore, the problem of correcting for phase errors can be transformed into maximizing the contrast of the obtained image, so the optimization algorithm of the maximum image contrast can be used to optimize the SAR image after digital beamforming, which can estimate the phase deviation between channels.

The contrast of the image can be defined as the ratio of the standard deviation of the image intensity to the mean value, which can be expressed as

$$f(\overrightarrow{\Delta\phi_n}) = \frac{\sqrt{E\left(\left[I_{DBF}^{\,2}(\overrightarrow{\Delta\phi_n}) - E\left(I_{DBF}^{\,2}\left(\overrightarrow{\Delta\phi_n}\right)\right)\right]^2\right)}}{E\left(I_{DBF}^{\,2}\left(\overrightarrow{\Delta\phi_n}\right)\right)} \tag{16}$$

where $I_{DBF}\left(\overrightarrow{\Delta\phi_n}\right)$ denotes the image after DBF with phase error and $E(\bullet)$ denotes the mean value. For the given SAR image, $E(\bullet)$ is a constant, so a simpler equivalent form of contrast can be obtained. Therefore, we choose a metric based on the 4-norm as the contrast function, where the p-norm could be expressed as

$$\|x\|_p = \sqrt[p]{\sum_{i=1}^{N}|x_i|^p} \tag{17}$$

So, our objective function eventually can be defined as

$$f(\overrightarrow{\Delta\phi_n}) = \left(\left\|I_{DBF}\left(\overrightarrow{\Delta\phi_n}\right)\right\|_4\right)^4 = \sum_{i,j}\left|I_{DBF}\left(\overrightarrow{\Delta\phi_n}\right)\right|^4 \tag{18}$$

where *i, j* is the pixel coordinate of the image. Therefore, the estimated phase error is

$$[\Delta\phi_1, \Delta\phi_2, \ldots, \Delta\phi_{N-1}] = \arg\max_{\Delta\phi_1, \Delta\phi_2, \ldots, \Delta\phi_{N-1}} f\left(\overrightarrow{\Delta\phi_n}\right) \tag{19}$$

assuming that this average contrast is a smooth function of the calibration vector $\overrightarrow{\Delta\phi_n}$. Finally, the gradient descent method is used to continuously optimize the above equation with the following steps:

1.  Obtain the raw data of each channel;
2.  Calculate the gradient of the contrast function;
3.  Update the phase correction function

$$\overrightarrow{\Delta\phi_{n+1}} = \overrightarrow{\Delta\phi_n} - \mu\overrightarrow{\nabla_{\overrightarrow{\Delta\phi}}}f(\overrightarrow{\Delta\phi_n}) \tag{20}$$

4.  Determine whether the convergence is satisfied; if not, repeat step 2.

## 4. Results

In this section, simulation experiments are conducted using the raw data of the forward side-looking strip mode airborne SAR, and the echo data are used to simulate a 10-channel DBF-SAR system with the parameters shown in Table 1.

**Table 1.** Main parameter of the airborne SAR system and simulate 10-channel DBF-SAR.

| Parameter | Value |
|---|---|
| Platform height | 20 km |
| Platform velocity | 154 m/s |
| Carrier frequency | 9.6 GHz |
| Bandwidth | 480 MHz |
| Pulse duration | 2.4 μs |
| Antenna inclination angle | 33° |
| Antenna height | 1 m |
| Channel number | 10 |

To simulate the external calibration scene, three strong points are added artificially to the original echo in the azimuthal direction to simulate the corner reflector.

To simulate the DBF-SAR channels imbalance, the time delay, amplitude, and phase errors are added to each receive channel, except the reference channel, in the process of raw data generation for each channel as shown in Table 2, and Gaussian white noise is

added. The time delay is uniformly distributed in the interval of [−30 ns, 30 ns], the amplitude error is uniformly distributed in the interval of [−3 dB, 3 dB], the phase error is uniformly distributed in the interval of [−45°, 45°]. Let channel 1 be the reference channel to simulate the transmitted channel of the DBF-SAR.

**Table 2.** Preset value of amplitude, phase and time delay bias of multiple-channels SAR.

|  | **Channel 1** | **Channel 2** | **Channel 3** | **Channel 4** | **Channel 5** |
| --- | --- | --- | --- | --- | --- |
| sampling delay (ns) | 0 | 30 | −5.77 | −24.21 | 26.52 |
| amplitude error (dB) | 0 | −1.18 | 1.21 | 0.78 | −0.18 |
| phase error (°) | 0 | 26.53 | 12.99 | −10.93 | 28.04 |
|  | **Channel 6** | **Channel 7** | **Channel 8** | **Channel 9** | **Channel 10** |
| sampling delay (ns) | 20 | −22.08 | 27.37 | 4.51 | −15.91 |
| amplitude error (dB) | −0.15 | 0.89 | 0.56 | 1.37 | −2.79 |
| phase error (°) | 2.95 | −13.43 | 39.51 | 33.83 | 4.51 |

### 4.1. Sampling Time Delay Estimation

Since a strong point target in a SAR image is usually composed of multiple pixel points with high amplitude values, a certain bias will occur if we choose to directly compare the peak point locations in the image. Figure 5a shows the simulation of point target imaging before interpolation.

Here, we choose a sinc tenfold interpolation, and Figure 5b shows the simulated image of the point target after sinc interpolation.

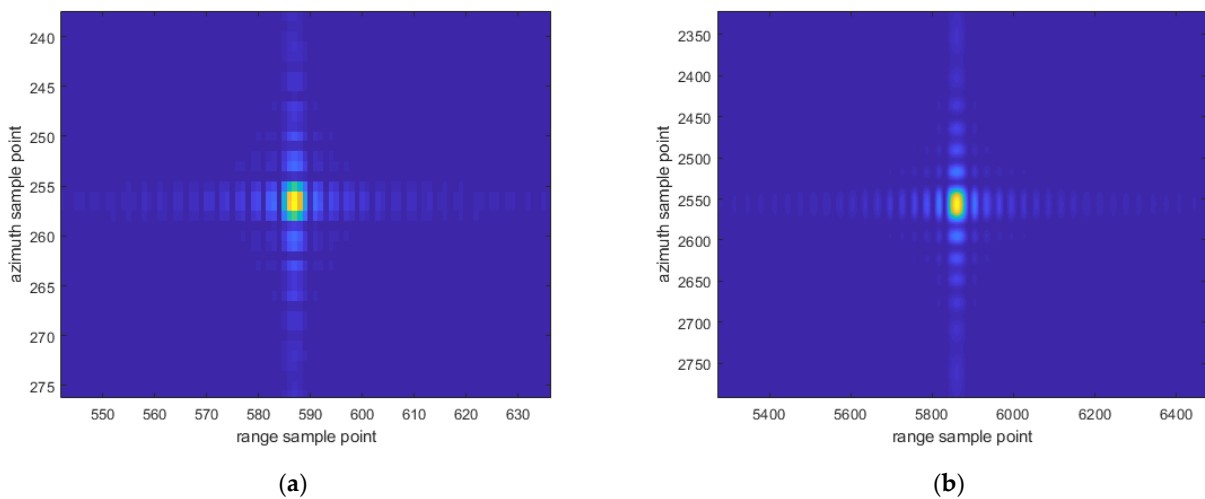

(**a**)                         (**b**)

**Figure 5.** Point target simulation. (**a**) Before interpolation; (**b**) after interpolation.

It can be seen as shown in Figure 5 that the point target image is smoother after tenfold interpolation. Therefore, by extracting the positions of the three strong point targets in the reference channel image and the error channel image and taking the statistical average of the deviations of the positions of all the strong point targets, we can obtain the estimated results, as shown in Table 3. We call the direct comparison of peak point locations without interpolation method 1 and then compare it with the method proposed in this paper.

**Table 3.** Time delay error estimation results of multi-channel DBF-SAR.

|  | **Channel 1** | **Channel 2** | **Channel 3** | **Channel 4** | **Channel 5** |
|---|---|---|---|---|---|
| setting value (ns) | 0 | 30 | −5.77 | −24.21 | 26.52 |
| method 1 | 0 | 31.08 | −5.02 | −25.35 | 27.83 |
| this paper method | 0 | 30.07 | −5.47 | −24.60 | 26.43 |
|  | **Channel 6** | **Channel 7** | **Channel 8** | **Channel 9** | **Channel 10** |
| setting value (ns) | 20 | −22.08 | 27.37 | 4.51 | −15.91 |
| method 1 | 20.89 | −21.02 | 26.53 | 5.38 | −16.87 |
| this paper method | 20.02 | −21.81 | 27.35 | 4.55 | −16.14 |

### 4.2. Amplitude Error Estimation

The sampling delay of each channel obtained from the above estimation is compensated, and the amplitude is estimated again for the channel again after compensating for the time delay error. In the absence of noise interference, the reference echo directly after pulse compression is compared with the amplitude maximum of the error echo due to the inclusion of strong point targets in the echo, which is the amplitude error. However, the actual work often cannot avoid the interference of noise, so we select each channel echo for imaging processing, which can enhance the gain. Figure 6 shows the local image of the reference image and an error image.

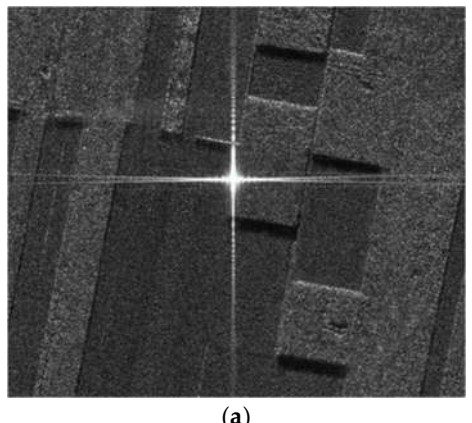 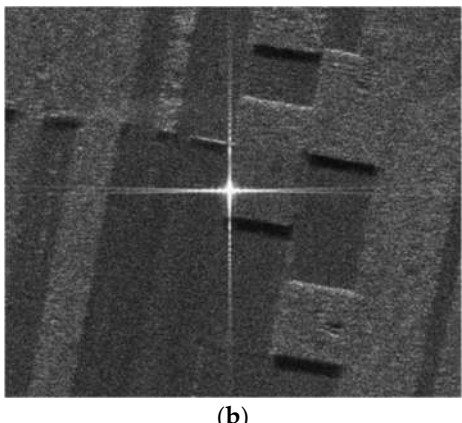

(**a**)          (**b**)

**Figure 6.** Absolute value local SAR image of a point target. (**a**) Reference channel absolute value SAR image; (**b**) imbalance channel absolute value SAR image.

Here, we call the traditional method of filtering noise on the satellite for each channel echo and then dividing directly by the absolute value as method 1 and compare it with the method in this paper. And the estimated results of method1 and this paper method are shown in Table 4.

**Table 4.** Amplitude bias estimation of multiple-channels SAR.

|  | **Channel 1** | **Channel 2** | **Channel 3** | **Channel 4** | **Channel 5** |
|---|---|---|---|---|---|
| setting value (dB) | 0 | −1.18 | 1.21 | 0.78 | −0.18 |
| method 1 | 0 | −1.04 | 1.03 | 0.95 | −0.83 |
| this paper's method | 0 | −1.16 | 1.19 | 0.76 | −0.17 |
|  | **Channel 6** | **Channel 7** | **Channel 8** | **Channel 9** | **Channel 10** |
| setting value (dB) | −0.15 | 0.89 | 0.56 | 1.37 | −2.79 |
| method 1 | −0.83 | 0.97 | 0.90 | 1.09 | −1.46 |
| this paper's method | −0.14 | 0.88 | 0.55 | 1.36 | −2.77 |

*4.3. Phase Error Estimation*

As can be seen from the estimated parameters obtained in Table 5, there is a large error in phase extraction of the reference channel image after comparing it with the error channel image. However, the optimized result obtained in this paper based on the maximum image contrast and continuous iteration by using the gradient descent method is very close to the setting value. We call the traditional method of filtering noise on the satellite for each channel echo and then extracting phase after dividing directly by the absolute value method 1, and then compare it with the method proposed in this paper.

**Table 5.** Phase bias estimation of multiple-channels SAR.

|  | **Channel 1** | **Channel 2** | **Channel 3** | **Channel 4** | **Channel 5** |
| --- | --- | --- | --- | --- | --- |
| setting value (°) | 0 | 26.53 | 12.99 | −10.93 | 28.04 |
| method 1 | 0 | 25.02 | 12.03 | −11.88 | 27.16 |
| this paper's method | 0 | 26.41 | 12.85 | −11.81 | 27.91 |
|  | **Channel 6** | **Channel 7** | **Channel 8** | **Channel 9** | **Channel 10** |
| setting value (°) | 2.95 | −13.43 | 39.51 | 33.83 | 4.51 |
| method 1 | 1.98 | −14.38 | 37.26 | 32.03 | 3.84 |
| this paper's method | 2.87 | −14.32 | 39.36 | 33.77 | 4.42 |

*4.4. Experiment Result Analysis*

Through simulating the raw data of the forward side-looking strip mode airborne SAR and the simulation data of three point targets, we could obtain the above experiment result.

As can be seen from Figure 7, the average delay estimation deviation of the delay error estimation method is 1.21 ns before interpolation, and 0.28 ns after tenfold interpolation. The average estimation error of the amplitude error estimation method is 0.43 dB compared with that of the echo directly. The average estimation error of the comparison after imaging is 0.02 dB. The phase error estimated method, which directly extracts the image after imaging, obtains 1.22° phase average estimation error, and maximizes the image contrast to obtain 0.28° phase average estimation error.

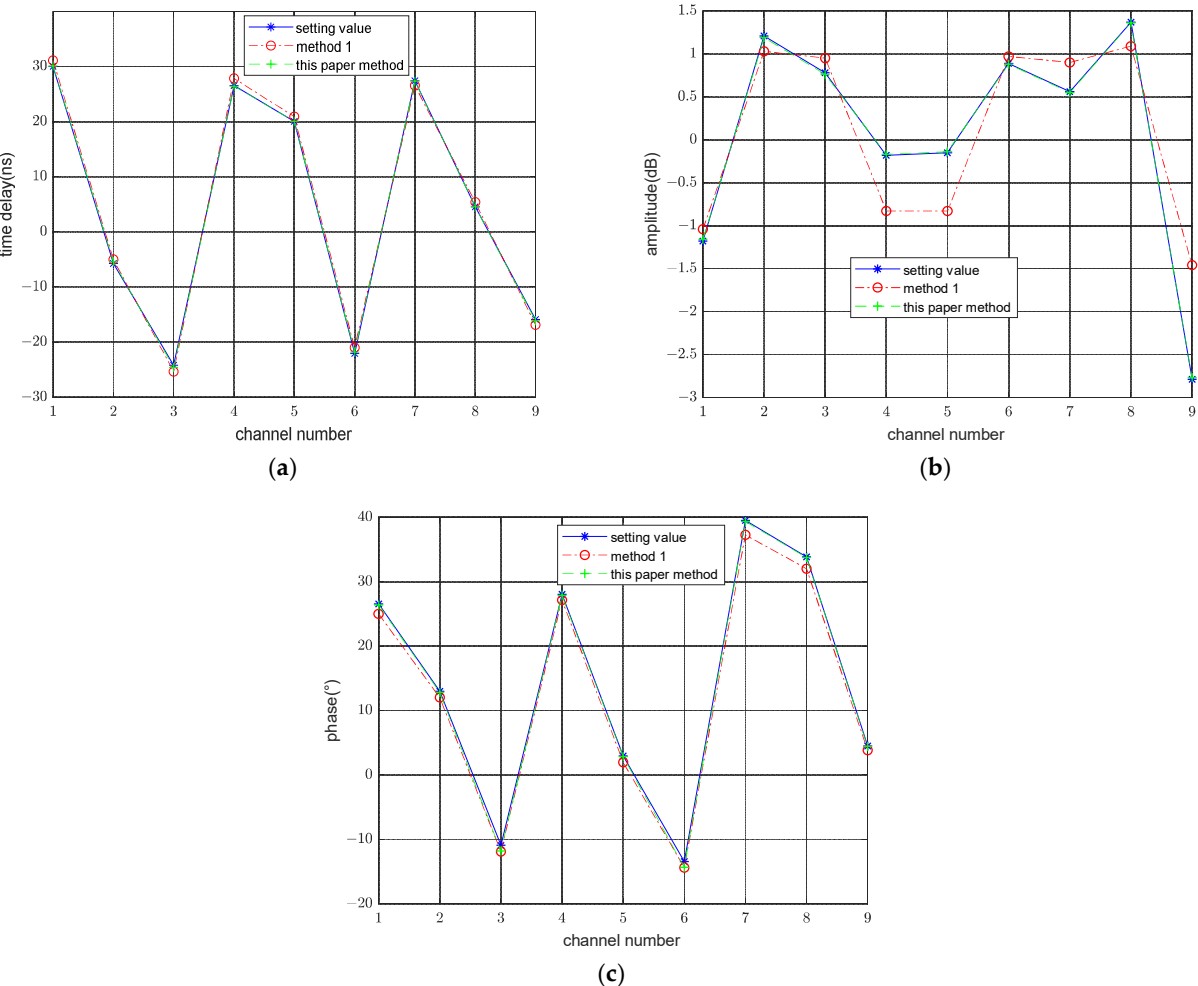

**Figure 7.** Comparison between the proposed method and previous methods. (**a**) Sampling time delay; (**b**) amplitude; (**c**) phase.

In order to verify the effect of the error estimation, we used the single-channel data, 10-channel DBF-SAR before calibration and 10-channel DBF-SAR after calibration. The calibration effect is obvious. Among them, Figure 8a is the result of single-channel imaging. Figure 8b,c are the results before and after calibration of 10-channel DBF-SAR. We selected the areas A (like A1, A2 and A3), B (like B1, B2 and B3) and C (like C1, C2 and C3) to observe the local details after DBF-SAR calibration as shown in Figure 8a–c. Figure 8d–f are the house. Figure 8g–i are the ground calibration points. Figure 8j–l are farmland. The whole scene and the corresponding local zoomed-in view for some special objects are compared and analyzed. The house, ground calibration points and the farmland in Figure 8f,i,l are clearer after calibration. Owing to the improvement of the SNR, it is obvious that the results after 10-channel DBF processing can provide rich detailed information about the feature space and prominent structure and texture in the wide swath observation mission.

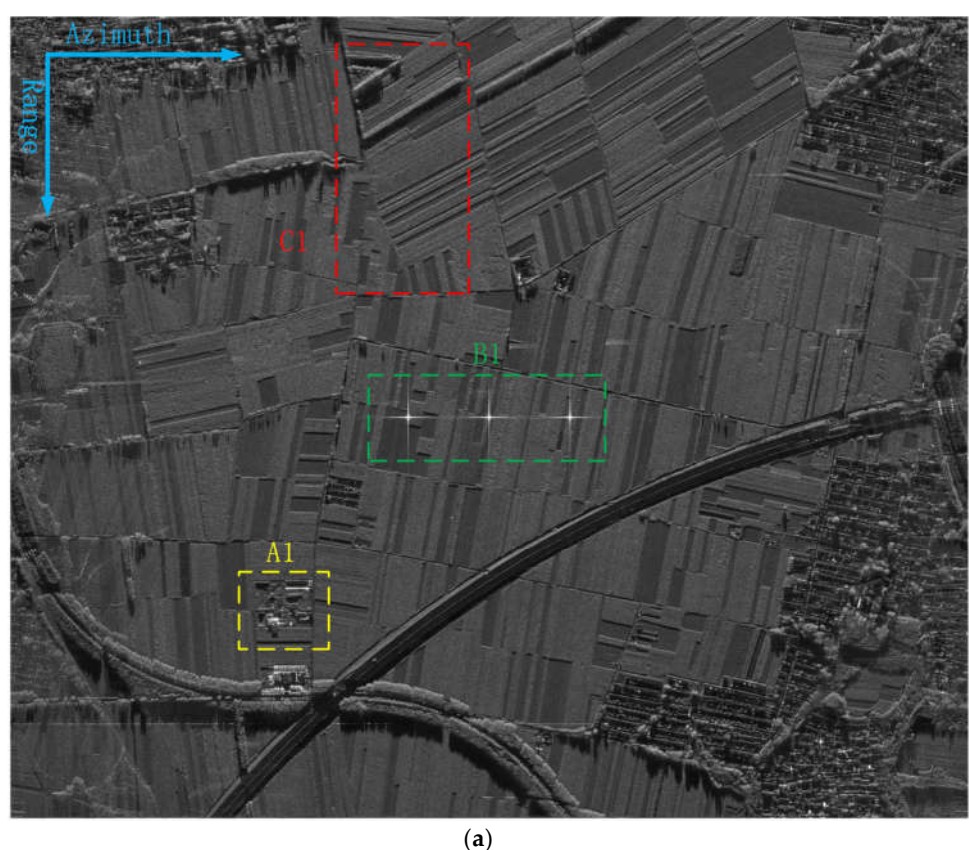

(**a**)

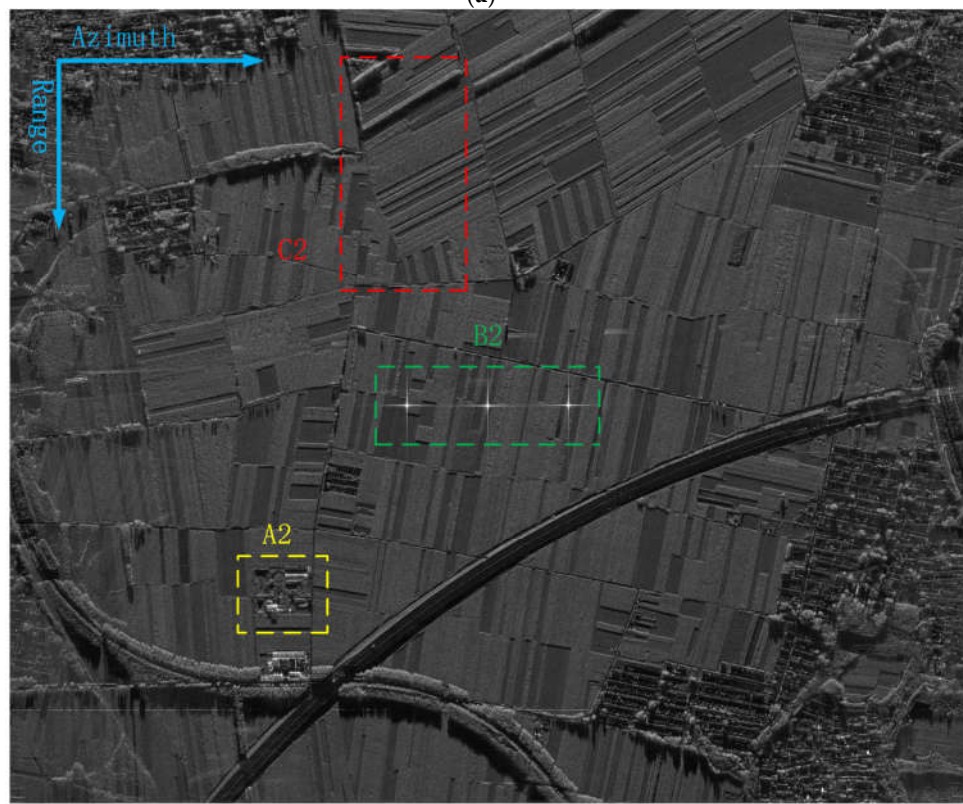

(**b**)

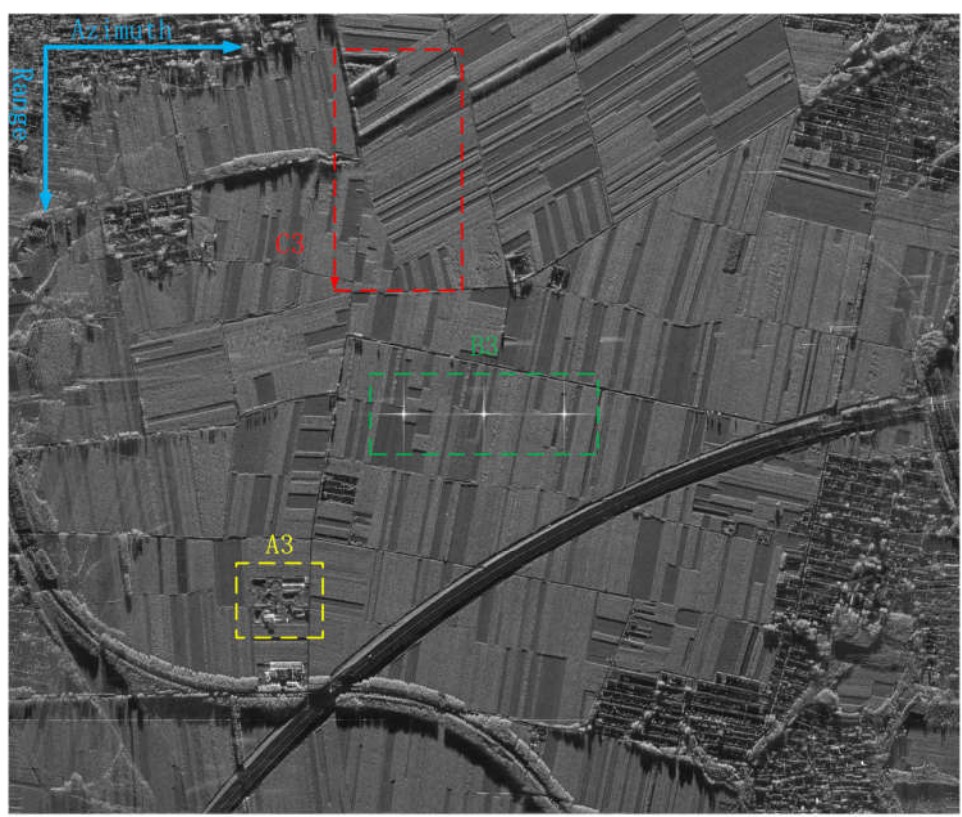

(**c**)

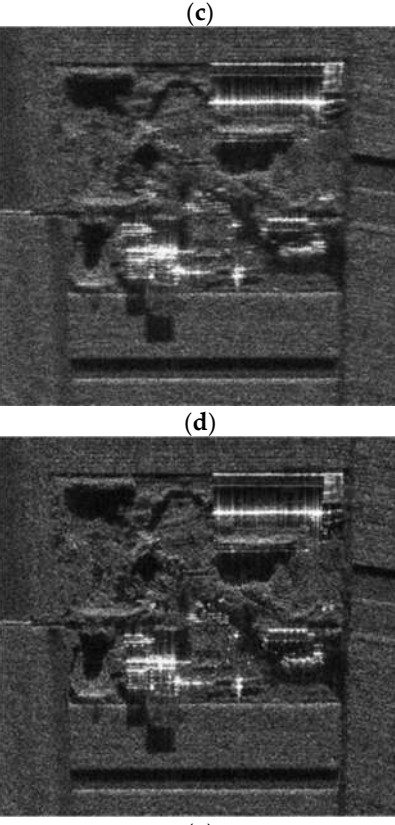

(**d**)

(**e**)

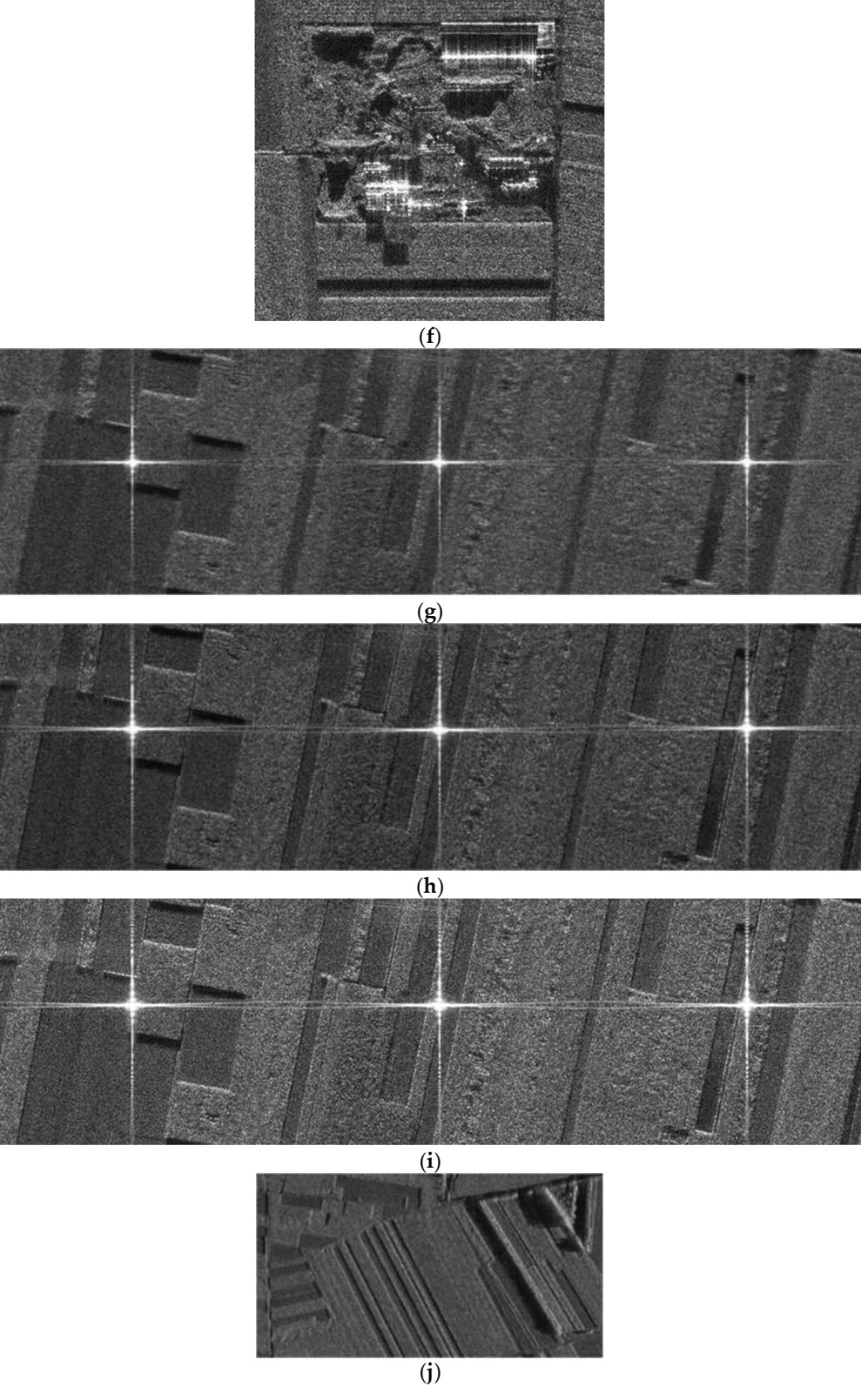

(**f**)

(**g**)

(**h**)

(**i**)

(**j**)

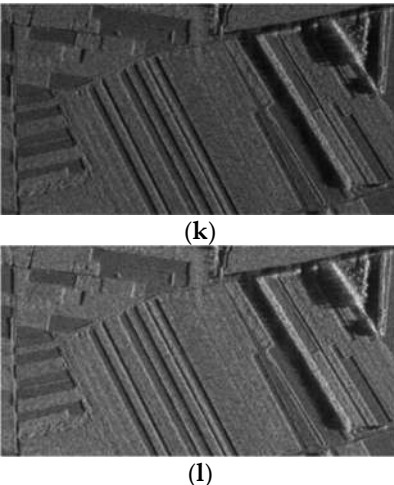

**Figure 8.** (**a**) Result of single-channel imaging: (**b**,**c**) results before and after calibration of 10-channel DBF-SAR; (**d–f**) The house; (**g–i**) The ground calibration points; and (**j–l**) the farmland.

The DBF-SAR could form high gain and narrow beam to receive echo to obtain the high gain SAR image after digital beamforming. As can be seen from Figure 8, compared with the single-channel SAR image, the SAR image after digital beamforming has a great improvement in the signal-to-noise ratio. As can be seen from Figure 8b,c, the delay, amplitude and phase errors are corrected by the channel calibration method proposed in this paper, and the quality of the DBF-SAR image after compensating has improved significantly. The SNR of each image in Figure 8 can be obtained as shown in Table 6.

In Table 6, after channel error correction, the SNR of SAR image after digital beamforming with channel error correction improves by about 5 dB, compared with that of the image after digital beamforming without channel error correction, and by about 8 dB compared with that of the single-channel SAR image, the channel signal coherence between the obvious ascension. Through comparing the A, B and C areas of the three images, it can be seen that, compared with the single channel imaging result, the DBF processing signal gain without the channel calibration processing is higher. Compared with the results without channel equalization, the DBF processing signal with channel calibration energy relative to the noise is enhanced obviously, and the image detail is more obvious. Finally, combined with Table 6 and Figure 8, the effectiveness of the proposed method is verified.

**Table 6.** SNR of single channel, DBF-SAR without channel calibration and DBF-SAR with channel calibration.

|  | Single Channel | DBF-SAR without Channel Calibration | DBF-SAR with Channel Calibration |
|---|---|---|---|
| SNR (dB) | 20.55 | 23.67 | 28.04 |

## 5. Discussion

According to the above experimental results, the DBF-SAR channel calibration method based on external calibration is practical and feasible. It can accurately estimate the time delay, amplitude and phase error between channels using external calibration points. Among them, by calculating the average delay estimation error of each channel, we can conclude that the estimation accuracy can reach 0.28 ns, and its performance is 0.4 mm accuracy at the slant range. Its estimation accuracy meets the requirement. Calculating the average amplitude estimation error of each channel, the estimation accuracy can reach 0.02 dB, and the estimation result is very accurate. The average phase estimation error of each channel can be calculated, and the estimation accuracy can reach 0.28°, which can also meet the needs in engineering. Moreover, by comparing the 10-channel DBF-SAR

image without channel equalization with the single-channel SAR imaging without DBF, it can be seen that the DBF technique can effectively improve the signal-to-noise ratio and gain of the image.

Similarly, for each receive channel containing error after calibration, the imaging effect is greatly improved when DBF imaging is performed. Based on the above experimental results, our proposed external calibration method for DBF-SAR is practical and effective and can be applied to the actual calibration project. On account of no spaceborne DBF-SAR in orbit at present, there is no way to verify the compensation effect in the actual situation with specific projects. Therefore, the future study needs to be combined with specific calibration requirements to verify the role in the actual engineering.

## 6. Conclusions

DBF-SAR can effectively resolve the contradiction between azimuth resXolution and pitch direction mapping width by using multiple channels in the elevation to achieve high resolution and wide swath. However, the imbalance between the multiple channels in the elevation of DBF-SAR will lead to the degradation of the radar imaging performance after digital beamforming. For the channel imbalance problem, the traditional idea is to use internal calibration loops for correction and then further correction in the echo data domain. However, for multi-channel systems, especially DBF-SAR systems, the number of channels in the elevation is usually numerous, channel calibration using the calibration loop is challenging in hardware, and the complex structure on the satellite makes it difficult to estimate and correct the echoes received from each channel on the satellite in real time. In this paper, we propose a method to estimate the time delay, amplitude and phase error based on the external calibration data. The strong scattering property of the corner reflector is sufficiently utilized to achieve the external calibration. Finally, by comparing the estimated value with the preset value and simulating the single-channel image, the 10-channel DBF-SAR image without channel equalization and the 10-channel DBF-SAR image after channel equalization, the accuracy of the estimation is verified to meet the engineering requirement, and the validity of the method is verified.

**Author Contributions:** Conceptualization, H.C., F.M. and L.L.; methodology, H.C. and F.M.; validation, G.L.; formal analysis, H.C. and F.M.; investigation, F.M. and L.L.; resources, F.M. and L.L.; data curation, G.L.; writing—original draft preparation, H.C.; writing—review and editing, H.C., F.M. and L.L.; All authors have read and agreed to the published version of the manuscript.

**Funding:** This research was funded by the National Natural Science Foundation of China (No. 61571417).

**Data Availability Statement:** The data presented in this study are available on request from the corresponding author. The data are not publicly available due to restrictions of privacy.

**Conflicts of Interest:** The authors declare no conflict of interest.

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
