# Peer review of "Elevation Multi-Channel Imbalance Calibration Method of Digital Beamforming Synthetic Aperture Radar"

_remotesensing, doi:10.3390/rs14174350_

Round 1

Reviewer 1 Report

Dear Authors,

Thank you very much for the article. I think, the article truly advances scientific knowledge and has a very high practical value. I found that almost all the concepts are presented clearly and that the article is prepared at a high level. The article is definitely worth publishing, in my opinion.

Please see the specific comments below:

11.      Lines 28-29. “Synthetic Aperture Radar (SAR)[1] is an active ground sensing system which could  not be affected by clouds, rain, fog and other climatic conditions.

This is not exactly true. Atmosphere affects SAR, especially working on shorter wavelengths. The comprehensive reading regarding the issue is for example Skolnik, Radar handbook. There are also issues and methods to correct the atmospheric effects in InSAR. Therefore I suggest replacing the “could not be affected” to something like “might be used in the conditions..” or similar.

22.      Line 165, equation 5. In the upper part of the equation (5), please write R2e with “e” directly below “2” (squared) rather than further to right.

33.      Line 184, “tiny”. Please add numerical values/estimations, references if possible, and in general elaborate on this. This is a very important statement for the whole following methodology, so it should be described better.

44.      Line 188. The equation (9) number sits alone on the line. Please move it to the same line as the equation itself, if possible.

55.       Line 204-205 “which can significantly reduce the amount of in-orbit computing.”. Will it increase the amount of the transmitted data?

66.      Figure 4. On the left part of the image, the central block is named “Estimated the error”. It seems that “the” is unnecessary. It also might be “Error estimation” or whatever. Please check.

77.      Please describe your corner reflector. Dihedral, trihedral? Size?

88.      Please start paragraphs 3.2 and 3.3. from capital letters.

99.      Line 243, “place”. Should it be “placed”?

110.   Line 243 and all the text. Please check double-spaces and spaces between words and commas throughout the text.

111.   Lines 295-300. There is some confusion with the numbers of steps. 1,2, then 1,2 again. Please make it 1,2,3,4, or 1,2,2.1.,2.2.

112.   Table 1. What is the platform, which carries the radar? Drone?

113. Figure 7 caption. Please add spaces between brackets and words.

   14. Line 369. “selecte”. Should it be “selected”?

1 15. Fig. 8 a,b,c. Letters and numbers (for example “A2”, “Range”) have very low resolution. Please improve.

1 16. Line 394. “Finally” starts from the capital letter after a comma.

Thank you very much and good luck with the publishing of the article and your future research!

Kind Regards

Your Reviewer.

Reviewer 2 Report

The manuscript presents a method to estimate the sampling time delay error, amplitude error and phase error based on the external calibration data for DBF-SAR. The processing framework has been tested through simulated airborne SAR data. The novelty is new and applicable, very detailed experiments are provided. The reviewer suggests its acceptance with minor edits.

Comments to the Author

1)        The use of English in this paper needs to be moderately improved. All grammar and typing errors need to be corrected (e.g., line41-43 in page 1, line162-164 in page 5).

2)        The method proposed in this paper requires corner reflectors. What if there are no corner reflectors?

3)        In line 216-218 in page 7, what are (x_(ref,i),y_(ref,i)) and (x_(n,i),y_(n,i))?

4)        What does method 1 in Table 3 refer to? What is the specific process?
